# Tumor Evolution and Therapeutic Choice Seen through a Prism of Circulating Tumor Cell Genomic Instability

**DOI:** 10.3390/cells10020337

**Published:** 2021-02-05

**Authors:** Tala Tayoun, Marianne Oulhen, Agathe Aberlenc, Françoise Farace, Patrycja Pawlikowska

**Affiliations:** 1Gustave Roussy, Université Paris-Saclay, “Circulating Tumor Cells” Translational Platform, CNRS UMS3655–INSERM US23AMMICA, F-94805 Villejuif, France; tala.tayoun@gustaveroussy.fr (T.T.); marianne.oulhen@gustaveroussy.fr (M.O.); agathe.aberlenc@gustaveroussy.fr (A.A.); 2Gustave Roussy, INSERM, U981 “Molecular Predictors and New Targets in Oncology”, F-94805 Villejuif, France; patrycjamarta.pawlikowska@gustaveroussy.fr; 3Faculty of Medicine, Université Paris-Saclay, F-94270 Le Kremlin-Bicetre, France

**Keywords:** circulating tumor cells, genomic instability, chromosomal instability, DNA-repair, tumor genetic heterogeneity

## Abstract

Circulating tumor cells (CTCs) provide an accessible tool for investigating tumor heterogeneity and cell populations with metastatic potential. Although an in-depth molecular investigation is limited by the extremely low CTC count in circulation, significant progress has been made recently in single-cell analytical processes. Indeed, CTC monitoring through molecular and functional characterization may provide an understanding of genomic instability (GI) molecular mechanisms, which contribute to tumor evolution and emergence of resistant clones. In this review, we discuss the sources and consequences of GI seen through single-cell analysis of CTCs in different types of tumors. We present a detailed overview of chromosomal instability (CIN) in CTCs assessed by fluorescence in situ hybridization (FISH), and we reveal utility of CTC single-cell sequencing in identifying copy number alterations (CNA) oncogenic drivers. We highlight the role of CIN in CTC-driven metastatic progression and acquired resistance, and we comment on the technical obstacles and challenges encountered during single CTC analysis. We focus on the DNA damage response and depict DNA-repair-related dynamic biomarkers reported to date in CTCs and their role in predicting response to genotoxic treatment. In summary, the suggested relationship between genomic aberrations in CTCs and prognosis strongly supports the potential utility of GI monitoring in CTCs in clinical risk assessment and therapeutic choice.

## 1. Introduction

Circulating tumor cells (CTC), present in peripheral blood of patients with cancers, are released from spatially distinct metastatic sites and primary tumor and thus may provide a comprehensive genomic picture of tumor content. The number of CTCs consists an independent prognostic factor and can be used to monitor treatment efficacy [1,2]. Alongside technological advances, CTCs have attracted clinical interest as a liquid biopsy to detect predictive biomarkers of sensitivity and resistance for therapy selection. Moreover, recent data on single CTC genomic analysis revealed the wide heterogeneity of CTCs, emphasizing the potential clinical utility of single CTC sequencing in identifying resistant clones that are arguably an important subset of cancer cells to target and eradicate. Indeed, growing evidence shows that CTCs may represent tumor phenotypic, genomic and transcriptomic heterogeneity and hence constitute a valuable sample to investigate tumor vulnerabilities. The phenotypes associated with tumor resistance and metastases require a complex pattern of cooperating processes among which genomic instability (GI) is a major actor. Oncogenic mutations as well as large-scale genomic alterations, copy number changes, DNA damage repair deficiencies or cell cycle perturbations may serve as an origin of GI and subsequent tumor heterogeneity. By offering real-time monitoring of a constantly evolving disease and by examining tumor GI through simple blood draws, CTCs may be of great utility to monitor patient response to treatment and precision medicine. Moreover, CTC-derived models have recently emerged as tractable platforms to explore functional capacities of CTCs.

In this review, we discuss different sources of GI and their impact on potential therapeutic solutions. We explore CTC genomic heterogeneity through fluorescence in situ hybridization (FISH) and single-cell sequencing and discuss how profiling of CTCs can be used to trace GI of tumors. We emphasize the importance of GI characterization in the context of tumor evolution and therapeutic choice. We outline the availability and utility of CDX models in functional characterization of tumor-adapted GI mechanisms. Finally, we highlight the dynamic changes of DNA-repair-related protein expression as functional biomarkers of GI and/or response to genotoxic treatment. 

## 2. Genomic Instability, More Than a Hallmark of Cancer

Over the past few years, genomic studies have demonstrated the complex and heterogeneous landscape of cancer and its potential impact on treatment resistance and metastasis development. GI is a driving force promoting continuous modification of tumor genomes and leading to clonal evolution and tumor genomic heterogeneity. Alterations in the DNA damage response (DDR), endogenous and oncogene-induced replication stress or cell division deregulation promote GI in cancer (Figure 1).

### 2.1. DNA Damage Defects

The DNA damage response (DDR) is a dynamic process based on the successive recruitments of different actors to DNA lesions. DNA damage occurs as a result of exogenous events such as ionizing irradiation or intercross-link agents, or as a part of perturbed physiological processes (see “Replicative stress” below). Resulting DNA double-strand breaks (DSBs) are the most cytotoxic lesions. Typically, two main repair mechanisms intervene to repair DSBs: homologous recombination (HR) and classical nonhomologous end joining. Histone H2AX (γH2AX), Nijmegen breakage syndrome 1 (nibrin/NBS1) and mediator of DNA damage checkpoint protein 1 (MDC1) create a signal amplification loop adjacent to DSBs, which engages the recruitment of DDR proteins, including the MRN (MRE11-RAD50-NBS1) complex and breast cancer 1 (BRCA1) [3,4]. In-depth investigation of functional, “real time” biomarkers of DDR is crucial for monitoring this process under therapy. Phosphorylated γH2AX has emerged as a biomarker of DSBs, allowing the monitoring of genotoxic events [5]. Its expression also correlated with sensitivity to chemotherapy, radiotherapy, treatment with poly(ADP-ribose) polymerase (PARP) inhibitors (PARPi) and chemical genotoxicity [6,7]. 

Tumors deficient in one DNA repair pathway often rely on a compensatory mechanism to resolve the damage, i.e., fit their DNA-repair machinery, giving concomitantly potential opportunities for targeted therapeutic approaches. PARPi have demonstrated synthetic lethality in HR deficient BRCA1/BRCA2 mutant tumors, which led to their approval in platinum-sensitive (with/without BRCA1/2 mutation) ovarian cancer and in germline BRCA1/2 (gBRCA)-mutated metastatic breast cancer [8,9,10]. Germline g*BRCA* mutations remain the most common clinical biomarker for PARPi therapy response because BRCA-mutant cells show clear evidence of HR deficiency. The prevalence and clinical relevance of somatic mutations in Fanconi anemia (FA) genes (23 FANC genes identified up to now) have been recently reported as “BRCAness”, traits of sensitivity to PARPi treatment first identified in breast cancer and later acknowledged in other types of cancers [11]. Indeed, FA genes are commonly altered in several cancers. According to The Cancer Genome Atlas, alterations in FA genes (mutations, deletions, and amplifications) were detected in 40% of tumors [12]. The canonical function of FA proteins is to eliminate chromosome-breaking effect of intercross-linking agents and preserve genomic integrity by stabilizing replication forks, moderating RS and regulating mitotic division. Thus “BRCAness”-positive tumors are also frequently sensitive to platinum salts. However, amplifications of FA genes may be advantageous to cancer cells and contribute to resistance to chemotherapy. Deep deletions and loss-of-function mutations in DNA-repair-related genes may confer tumor sensitivity to DNA-repair-related targeted therapy. Recently, the potential utility of RAD51 protein, a surrogate marker of HR functionality, has been reported [13,14]. RAD51 assay performed in clinical practice on tumor tissue samples may improve patient selection for PARPi therapy in non-BRCA1/2-related cancers, which likewise present HR deficiency. 

### 2.2. Replicative Stress

Any possible obstacle that disturbs DNA replication and prevents cells from finalizing their genome duplication before mitosis causes replicative stress (RS). It is a frequent phenomenon among cancer cells and is usually associated with structural chromosomal instability (CIN), which arises from prone to damage under-replicated DNA. Many cancers harbor persistent RS due to oncogene activation or compromised DNA-repair machinery in the absence or loss-of-function of essential that ensure protection or repair of stressed replication forks. Indeed, constitutive activation of oncogenes such as c-MYC, HRAS and KRAS has been shown to disturb the accurate DNA replication and has been associated with increased GI [15,16,17]. Recently, Wilhelm et al. proposed a mechanism through which RS contributed to numerical aneuploidy in both healthy and CIN^+^ cancer cells, by driving chromosome mis-segregation via premature centriole disengagement [18]. This study was concordant with previously published observations where RS increased incidence of lagging chromosomes during cellular division [19,20]. Nonetheless, cancer cells cope with RS through different mechanisms, such as overexpression of checkpoint mediators Claspin and Timeless (members of ATR/CHK1 pathway), which may increase RS tolerance by protecting replication forks [21]. Therefore, similarly to DNA-repair-deficient tumors, RS response may also be exploited for cancer treatment. 

### 2.3. Cell Division Abnormality

Mitotic CIN is defined as inability to faithfully segregate equal chromosome contents to two daughter cells during mitosis. Indeed, abnormal chromosome numbers or numerical aneuploidy is a common alteration in human cancer. It may be promoted by mitotic checkpoint deregulation and may lead to the loss of tumor suppressors or gain of oncogenic signals. However, the loss of key mitotic checkpoint genes is rare in clinical samples. Whole-genome doubling (WGD) induced through cytokinesis failure is a one-off event which may promote aneuploidy. Its prognostic utility has been first shown in early-stage colorectal cancer and was later proposed in other cancer types [22,23]. Tumor cells experiencing WGD have developed centrosome clustering as a mechanism to prevent lethal mitotic spindle multipolarity, by merging multiple centrosomes into two functional spindle poles. Interestingly, centrosome amplification stimulates cytoskeleton alterations, which might in turn be responsible for tumor cell invasions and thus metastatic development [24]. Inhibition of centrosome clustering may represent an anti-tumor specific strategy based on the formation of multipolar spindles and subsequent tumor cell death [25]. GI has also been associated with epithelial-mesenchymal transition (EMT) through the activation of the cytosolic DNA response pathway [26]. Indeed, altered chromosome segregation arising from GI promotes micronuclei formation whose rupture spills DNA into the cytosol. Presence of DNA in the cytosol induces the cGAS-STING (cyclic GMP-AMP synthase-stimulator of interferon genes) cytosolic DNA-sensing pathway and downstream noncanonical NF-κB signaling, thus inducing a proinflammatory response, which factors were recognized as EMT stimulators [27]. Identification of cGAS/STING activators is an area of active research, with several ongoing clinical trials evaluating such molecules [28,29]. 

Sequencing studies and mechanistic investigations have revealed alterations in GI-related genes and events (e.g., *TP53*, *BRCA1/2, RB1* loss, *CDKN2A* loss) relevant in cancer progression [12,30]. These have important clinical implications as they may give the possibility to better stratify the patients and help clinicians in therapy selection. 

## 3. GI-Related Biomarkers in CTCs and Their Utility for Clinical Decision Making

In-depth assessment of GI in bulk biopsy sample is frequently incomplete due to limited sample availability, surrounding normal tissue contamination and tumor heterogeneity. Additionally, serial tumor tissue biopsies are not feasible in clinical practice and metastasis biopsies are limited to accessible sites. Blood-based liquid biopsies containing CTCs have emerged as a noninvasive and accessible alternative enabling serial sampling. CTC analysis is technically challenging due to their low prevalence in the bloodstream and their phenotypic heterogeneity. Nevertheless, several groups have recently illustrated the feasibility of single-cell profiling in CTCs, providing a spectrum of genomic alterations that may potentially represent tumor heterogeneity and unravel aggressive subclones. CTCs acquiring genomic alterations can initiate and drive selection of resistant clones responsible for tumor evolution and metastatic progression [31]. 

### 3.1. CIN Analysis in CTCs by FISH 

FISH technique has been adopted as one of the main methods for the assessment of CIN status in tumors (reviewed by McGranahan et al. [32]). Variations in chromosome copy number across the cell population can be quantified using fluorescently labeled DNA probes that bind to the centromeres of specific chromosomes. In CTCs, FISH has been developed and optimized to detect biomarkers of sensitivity to selected treatments and better stratify the patients. However, research revealed an unforeseen aspect of chromosomal heterogeneity across CTCs. Indeed, one of the first successful applications of the FISH assay showed important CIN in prostate cancer (PCa) CTCs through the detection of heterogeneous chromosomal abnormalities among patients [33]. A study in castration-resistance prostate cancer (CRPC) showed that *ERG* oncogene status was maintained in CTCs, while significant genetic heterogeneity was observed in *AR* copy number gain and *PTEN* loss. This suggested that *ERG* rearrangements might constitute an early event in prostate tumorigenesis [34]. In the multicentric PETRUS study of biomarker assessment, we reported phenotypic and FISH genetic heterogeneity of metastatic tumor tissue and CTCs in patients with CRPC [35]. High concordance between metastatic biopsies and CTCs for *ERG*-rearrangement was observed in spite of higher heterogeneity in CTCs. Other groups have also performed FISH analysis in metastatic CRPC CTCs revealing amplification of the *AR* locus and *MYC* [36] as well as the presence of PCa-specific TMPRSS2-ERG fusion [37]. The comparative detection of *ALK*-rearranged CTCs in NSCLC patients and corresponding tumor tissue biopsies was also performed. In a cohort of 87 patients with lung adenocarcinoma, positive ALK immunostaining was reported in CTCs isolated from five patients, corresponding to the same patients presenting *ALK*-rearranged tumors [38]. Our group reported the detection of unique *ALK* rearrangement patterns in CTCs in patients with metastatic NSCLC. Notably, we noted a high concordance in *ALK* rearrangement patterns between CTCs and tumor biopsies in 18 *ALK*-positive and 14 *ALK*-negative patients. Additionally, the presence of a unique *ALK* rearrangement pattern and EMT features was observed in CTCs [39]. Utility of *ALK* FISH testing in CTCs in the longitudinal follow-up of crizotinib resistance profiling was also demonstrated [40]. We showed that patients monitored at the early stage of crizotinib treatment presented significant correlation between dynamic evolution of the amount of *ALK* copy number gained in CTCs and PFS, suggesting that increased CIN in CTCs may be associated with a worse outcome in *ALK*-rearranged NSCLC [41]. These reports consistently demonstrate that monitoring tumor genomic characteristics via CTCs FISH analysis may serve as a predictive biomarker of treatment efficacy in NSCLC patients. 

In 2015, we reported the detection of rearrangement in the ROS1-tyrosine kinase gene (present in 1% of NSCLC) in CTCs from *ROS1*-rearranged NSCLC patients. High levels of aneuploidy and numerical CIN have been proposed as a mechanism of genetic diversity in CTCs of *ROS1*-rearranged patients. DNA content quantifications and chromosome enumeration underscored increased CIN in CTCs [42]. Further studies based on FISH analysis emphasized CTC genomic heterogeneity through assessment of their numerical CIN. Another report demonstrated the assessment of *MET* amplification by FISH in CTCs from *EGFR*-mutated NSCLC patients at progression on erlotinib. *MET* amplification was detected in 3 of 39 samples but interestingly all *MET*-amplified CTCs were identified at disease progression [43]. Similarly, *MET* amplification was detected using FISH technique in CTCs of patients with gastric, colorectal and renal cancers following a capture of c-MET-expressing cells [44]. This particular aberration may have prognostic importance if confirmed, as c-MET protein overexpression increases distinctly in metastasis [45]. 

In breast cancer, assessment of *HER2* status is considered as standard practice for therapy selection [46]. Interestingly, assessment of *HER2* amplification using FISH in CTCs has been reported by several groups and may be used to stratify patients eligible to HER2-targeted therapy [47,48,49]. *PTEN* gene loss may drive tumor progression through activation of PI3K/AKT pathway and occurs frequently in CRPC. *PTEN* gene status was assessed in CTCs using the Epic Sciences platform, which identifies CTCs through an algorithm-based image analysis followed by FISH [50,51]. *PTEN* losses determined by FISH in CTCs correlated with PTEN expression loss measured by IHC in corresponding tumors biopsies. They were also associated with worse prognosis in CRPC patients [50]. These FISH studies highlight the importance of serial CTC genomic analysis for the identification of biomarkers predictive of therapeutic efficacy in different cancer types. The data also emphasize heterogeneous CIN as a characteristic feature of CTCs from different tumor types and show the importance of single-cell analysis to evaluate CNA changes as possible mechanisms of resistance and/or tumor evolution. FISH analysis of tumor samples is in most cases still manually performed and is particularly laborious given the important number of hematopoietic cells still retained in enriched CTC fractions. Nevertheless, technological advancements in the field led to the development of semi-automated microscopy method that allows the identification of filtration-enriched CTCs from NSCLC and PCa patients and the detection of *ALK*, *ROS1* and *ERG* gains and rearrangements in these cells, as we reported (Figure 2) [52]. Moreover, integrated subtraction enrichment and immunostaining FISH (SE-iFISH) was used to characterize CTCs of patients with malignancies such as nasopharyngeal carcinoma or esophageal cancer. Notably, CTC karyotyping allowed the assessment of chromosome 8 aneuploidy, which strongly associated with chemotherapy efficacy and prognosis [53,54]. Aforementioned studies show that although FISH has been developed to detect biomarkers of sensitivity to different selected treatments, it constitutes a valuable tool for the assessment of CIN across CTCs.

### 3.2. Copy Number Alterations (CNA) Landscape to Describe CIN in CTCs

The rarity and biological heterogeneity of CTCs have imposed technical challenges for their isolation and analyses at the single-cell level and impacted the success of robust processing of complex and costly downstream methodologies. The single-nucleus next-generation sequencing relies on successful whole genome amplification (WGA) of an individual cell to generate good-quality DNA for subsequent sequencing. All WGA systems generate nonlinear amplification bias, which may decrease genome coverage and thus needs to be taken into consideration during sequence analysis [55]. Reproducible CNA patterns among single CTCs and corresponding metastatic biopsy were obtained after multiple annealing and looping-based amplification cycles of WGA of single CTCs from lung cancer patients [56]. Indeed, each CTC from an individual patient exhibited reproducible CNA patterns similar to the metastatic tumor but not the primary tumor. This report also showed that different patients with adenocarcinoma shared similar CNA patterns, whereas patients with small-cell lung cancer (SCLC) had distinctly different CNA patterns. CNA profiling studies in the context of GI suggested that certain genomic loci may confer a selective advantage for metastasis through their action on different signaling pathways. To tackle the issue of protocol speed for clinical applications, Ferrarini et al. developed a single-tube method consisting of a single step, with ligation-mediated PCR (LM-PCR) WGA for low-pass whole genome sequencing and CNA calling from single cells [57]. This was adapted to analyze CTCs from patients with lung adenocarcinoma and PCa. The *Ampli*1™ WGA-based low-pass workflow (Menarini Silicon Biosystems) successfully captured substantial heterogeneity across CTCs, highlighting the utility of single-cell profiling application for genome-informed therapeutic strategies [57]. Another group assessed GI through genome-wide copy number profiling of CTCs from seven metastatic CRPC patients [58]. CTCs were identified and characterized using the Epic Sciences CTC platform and subclonal tumor suppressor loss, oncogene amplification and GI were measured by the distribution of large-scale state transitions (LST) genome-wide (frequency of CNV breakpoints > 10 Mb). A broad range of copy number changes in *AR* and *PTEN* were detected in most CRPC patients accompanied by high heterogeneity in LST distribution, highlighting important GI in CTCs at the single-cell resolution [58]. Additional CNA profiling studies in CRPC highlight high levels of genomic heterogeneity among CTCs [59,60]. The compound losses of three tumor suppressors (*PTEN*, *RB1* and *TP53*) in PCa CTCs and the corresponding circulating tumor DNA analysis were recently reported and linked to the aggressive trait of the tumor [61]. Moreover, gains in *PTK2* and *MYC* together with *TP53* loss were also detected in CTCs and were strongly associated with poor prognosis in PCa patients. Despite frequent copy number traces that highly resembled corresponding biopsies, unique gains in *MYC* were revealed in CNA profiles of CTCs captured from apheresis of PCa patients [62]. Previously, *MYCN* gain and simultaneous *AR* loss was proposed as a possible mechanism of neuroendocrine differentiation in PCa tumor samples [63] and was later confirmed in CTCs as part of highly complex profile containing additional aberrations in *ERG*, *c-MET* and *PI3K* genes during CRPC progression [59]. Evaluation of CNA profiles in CTCs from metastatic breast cancer patients suggested potentially targetable alterations in *PTCH1* and *NOTCH1* that were absent in baseline biopsies, indicating subclonal tumor evolution [64]. The predictive value of CNA profiles of CTCs has also been recently evidenced in SCLC patients. Characteristic CNA signature of subsequent chemosensitivity was reported with an 83.3% accuracy to classify SCLC CTCs as chemosensitive or chemorefractory [65]. Similarly, predictive single CTC-based CNA score in the response to first-line chemotherapy was demonstrated in SCLC patients by Su et al. CNA profiles across CTCs of individual SCLC patients were highly concordant with copy number losses in two frequently inactivated genes, *TP53* and *RB1*, found in 64.6% and 81.3% of patients respectively [66].

Overall, single-cell heterogeneity revealed by CNA analysis clearly represents a challenge for CTC molecular biomarker studies. Nevertheless, in-depth analysis of a sufficient number of CTCs may allow the profiling of characteristic CNA burden, which may be informative for future treatment strategies. 

### 3.3. Using CTC-Derived Models to Investigate GI Mechanisms

Over the past decade, CTC-derived models have emerged as tractable tools to explore metastatic disease by studying the tumorigenic capacity of CTCs in several malignancies [67]. Despite technical challenges due to CTC rarity in the bloodstream, significant efforts were provided in the establishment of CTC-derived xenografts (CDX). The first one was generated in 2013 from breast cancer patient CTCs [68], while other groups reported successful models in lung, melanoma and prostate cancers [69,70,71,72]. We recently reported sequential acquisition of key genetic events promoting an aggressive neuroendocrine transformation in CRPC CDX. *PTEN* and *RB1* losses were acquired in CTCs, while *TP53* loss harbored in a subclone of the primary tumor was suggested as the driver of the metastatic event leading to CDX development. Interestingly, co-occurring losses of tumor suppressor genes *PTEN*, *RB1* and *TP53* were found in single CTCs characterized by extremely high CIN. Neuroendocrine transformation was promoted by the high number of CNAs and WGD, highlighting GI acquired during metastatic development [72]. In SCLC, single-cell analysis of CDX revealed the existence of co-existing heterogeneous cell subpopulations that are contributing to multiple concurrent resistance mechanism to chemotherapy [73]. Ex vivo expansion of viable CTCs has also been described [74,75,76,77,78]. Transcriptomic analysis of a CTC cell line derived from a metastatic colon cancer patient indicated altered expression of DNA-repair-related genes compared to a primary colon cancer cell line [77,79]. Another CTC-derived breast cancer cell line was recently established from a patient with metastatic estrogen receptor-positive breast cancer. Its CNA profile was highly concordant with that of patient CTCs and WES analysis deciphered alterations in common DNA damage-related genes (e.g., *ATM, CDKN1A*) [78].

The current time frame required for developing CTC-derived models does not allow for real-time monitoring of cancer patients and thus may not inform clinical decisions. However, their genomic analysis may help decipher molecular events involved in CTC-mediated tumor progression and reveal potential CTC biomarkers relevant for clinical management. 

### 3.4. DNA Repair-Related Protein Biomarkers in CTCs 

Functional analysis of DNA-repair-related protein expression in CTCs has been used as a pharmacodynamic biomarker for monitoring response to chemotherapy or targeted therapy (Table 1). Expression of DSB marker γH2AX has been evaluated as a dynamic indicator of DNA damage in CTCs from patients with advanced cancers after topotecan treatment using immunofluorescent staining followed by FACS analysis [80]. Data showed feasibility of monitoring dynamic changes in CTC nuclear biomarkers at response to treatment. γH2AX foci were also evaluated in CTCs after CellSearch analysis performed during radiation therapy as well as during combination treatment of low-dose of radiotherapy combined with PARPi [81,82]. Another DSB protein, RAD50, has been sequentially monitored in CTCs and its expression was estimated after radiotherapy of single side lesions in advanced lung cancer patients. CTCs were additionally screened for the immunotherapeutic target PD-L1 after enrichment with CellSieve Microfiltration Assay [83]. Results showed that RAD50 nuclear foci formation in CTCs may serve as a noninvasive tracer in cancer patients receiving side-directed radiotherapy independently of PD-L1 screening. ERCC excision repair 1 (*ERCC1*) is required for the repair of cisplatin-induced DNA lesions and may play the role of a biomarker for predicting response to platinum therapy. Indeed, it has been suggested that tumor cells overexpressing ERCC1 may be characterized with an enhanced capacity to resolve DNA platinum-adducts and consequently bypassing platinum cytotoxicity [84]. ERCC1 expression in CTCs was found to negatively correlate with PFS in metastatic NSCLC patients under platinum-based chemotherapy [85] and presence of CTCs expressing ERCC1 after therapy indicated a worse outcome for breast cancer patients [86]. Another group showed that ERCC1 transcript expression in CTCs was more predictive of response to platinum-based chemotherapy than standard ERCC1 protein expression detected on primary tumor biopsy samples [87]. Additionally, ERCC1 transcript-positive CTCs were used for monitoring platinum-based chemotherapy and to assess the post-therapeutic outcome of ovarian cancer [88]. These studies suggested that CTCs may represent dynamic intra-cellular changes in response to DNA-repair-related treatments more accurately than tumor biopsy. Furthermore, overexpression of the DNA/RNA helicase Schlafen family member 11 (SLFN11) has been described as an emerging biomarker of tumor cell sensitivity to DNA-damaging agents, including platinum chemotherapy [89] and to PARPi in several cancers [90,91]. SLFN11 protein expression was evaluated by immunofluorescent staining in CTCs from CRPC patients treated with platinum chemotherapy. SLFN11 overexpression in CTCs was associated with longer PFS compared to patients with *SLFN11*-negative CTCs [92]. Despite accumulating data, identification of CTC subpopulations expressing DNA-repair-related markers remains complex due to the existing variations among the technologies used to this end, as well as their low prevalence in patient blood. Therefore, further research is required to determine the clinical relevance of such biomarkers, notably in patients with advanced malignancies presenting significant levels of CTCs. 

## 4. Conclusions

The study of GI-related biomarkers in CTCs is an emerging field, and their real-time monitoring may be useful in clinical decision making. The technical advances and robust CTC isolation methods may now allow us to capture phenotypic and genetic heterogeneity and, subsequently, to reconstitute tumor characteristics. The relationship between GI, prognosis and acquired resistance to treatment is very complex, and deciphering the molecular mechanisms contributing to GI in CTCs remains crucial. The advancements in FISH analysis have strongly contributed to the unveiling of increased CIN in CTCs and its potential role in resistance mechanisms. CNAs successfully assessed via single-cell sequencing of CTCs indicated various sources of GI, such as oncogene-induced replicative stress, cell-cycle-related genes alterations or WGD, suggesting a rationale for therapeutic options. Moreover, CNA events reveal common DNA-repair-related gene alterations detected across tumor types. Those DDR alterations increase GI and thus may constitute novel therapeutic targets. Single CTC sequencing may therefore provide insight into the mechanistic origins and consequences of DDR deficiency in cancer (Figure 3). Finally, CTC-based monitoring of DDR-related biomarkers was proven to inform about therapeutic progress, but it also indicates first signals of acquiring resistance. Therefore, though investigating GI mechanisms through CTC monitoring is challenging, it is becoming particularly useful for tracking tumor heterogeneity and may present a critical element for precision medicine.

## Figures and Tables

**Figure 1 cells-10-00337-f001:**
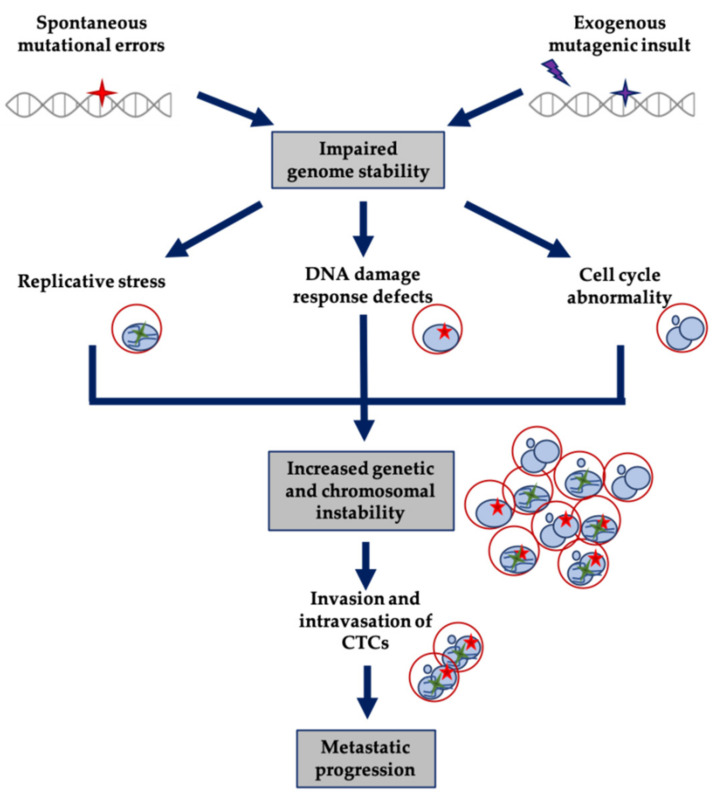
Concept diagram representing mechanisms of genome instability implicated in tumor evolution, including CTC contribution and their potential exploitation as biomarkers.

**Figure 2 cells-10-00337-f002:**
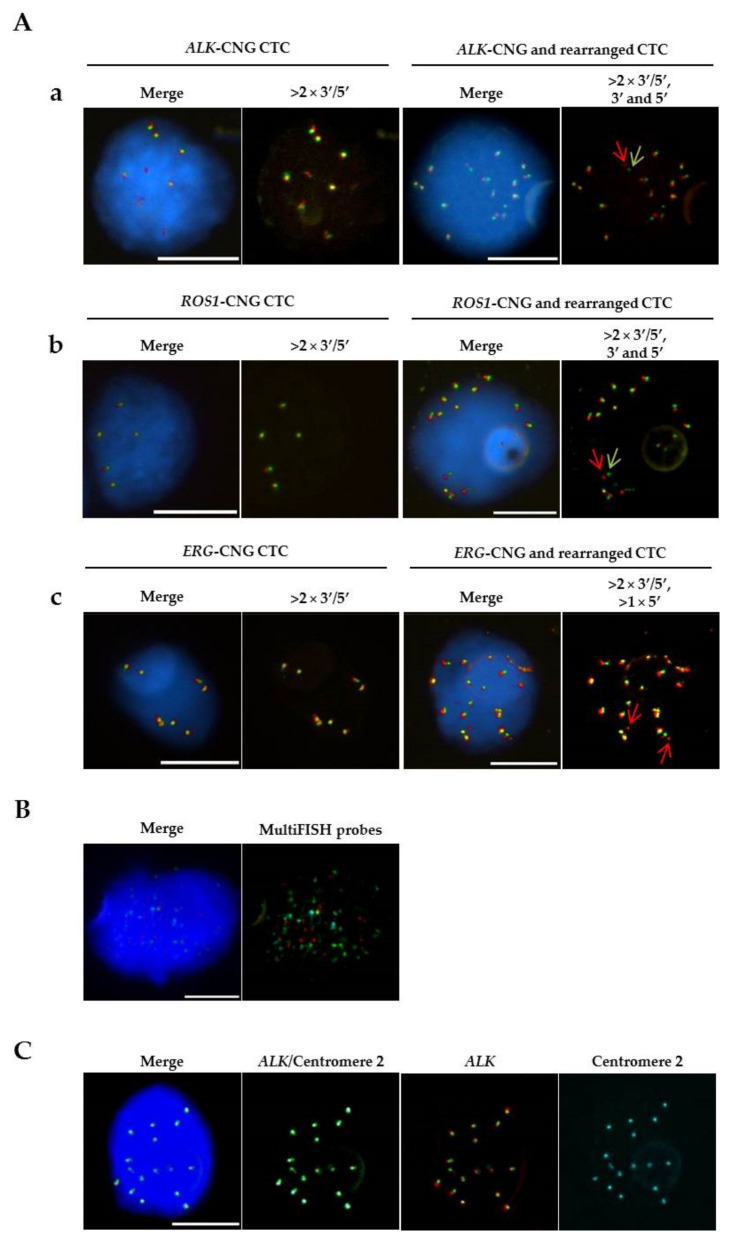
Detection of CTCs harboring *ALK* and *ROS-1* gene aberrations in NSCLC patients and *ERG* gene alterations in metastatic CRPC patients by combined immunofluorescent staining and filter-adapted FISH (FA-FISH). (**A**). (**a**) Example of FISH patterns in NSCLC CTCs with *ALK*-copy number gain (*ALK*-CNG) and *ALK*-rearrangement. Red and green arrows correspond to *ALK* 3′ and *ALK* 5′ probes (Vysis *ALK* Break Apart rearrangement Probe Kit from Abbott Molecular Inc., Chicago, IL, USA) respectively. (**b**) Example of FISH patterns in NSCLC CTCs bearing *ROS1*-CNG and *ROS1*-rearrangement. Green and red arrows correspond to 3′ and 5′ *ROS1*-rearrangement extremities (Vysis 6q22 *ROS1* Break Apart FISH probe RUO Kit from Abbott Molecular Inc.) respectively. (**c**) Example of FISH patterns in CRPC CTCs with *ERG*-CNG and *ERG*-rearrangement. Green and red arrows correspond to 3′ and 5′ *ERG* gene ends (Kreatech *ERG* Break Apart Rearrangement Probes kit) respectively. (**B**). Example of hybridized CTC using the AneuVysion Multicolor DNA Probe Kit (Abbott Molecular Inc.). Green spots indicate hybridization of locus-specific identification (LSI) 13 probe and centromere-specific enumeration probe (CEP) X. Red spots indicate hybridization of LSI 21 probe and CEP Y. Blue spots indicate hybridization of CEP 18. (**C**). Example of FISH patterns in CTCs with *ALK*-CNG detected by combined immunofluorescent staining and three-color FA-FISH for *ALK* gene and chromosome 2 centromere detection (XCyting Centromere Enumeration Probe XCE2 from MetaSystems GmbH), showing the existence of true gains of *ALK* gene in CTCs. Scale: white bars = 10µm.

**Figure 3 cells-10-00337-f003:**
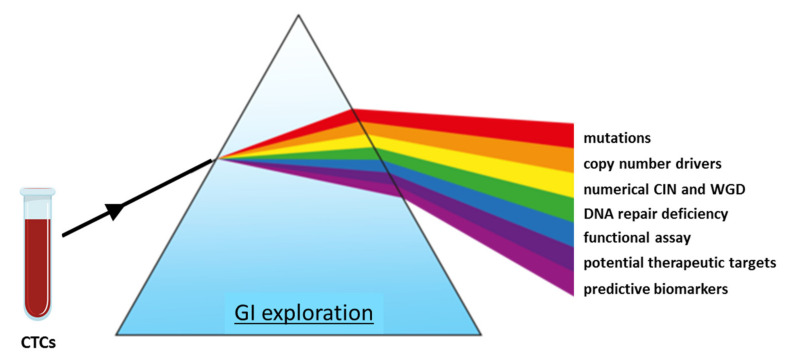
Schematic model of state-of-the-art strategies for the investigation of genome instability in CTCs.

**Table 1 cells-10-00337-t001:** DNA damage repair-related biomarkers in CTCs.

DNA Repair-Related Protein Markers in CTCs	Tumor Type	Treatment	Key Findings	Ref.
ϒH2AX (phosphorylated Ser 139 H2AX variant histone)	Various advanced cancers	Topotecan	- A dose-dependent increase of ϒH2AX-positive patient CTCs with topotecan - Monitoring of pharmacodynamics effects of chemotherapy via nuclear ϒH2AX levels	[80]
	NSCLC	Radiotherapy	Elevated ϒH2AX signal in CTCs post-radiotherapy	[81]
	Peritoneal cancers and advanced solid malignancies	Radiotherapy and PARPi (veliparib)	- Exploratory study showing the use of ϒH2AX in CTCs - Increase in ϒH2AX^+^ CTC levels after treatment in few patients while one patient presented a decrease, suggestive of treatment failure	[82]
RAD50 (double strand break repair protein)	NSCLC	Radiotherapy	- RAD50 foci formation used to label and track CTCs subjected to radiation at primary site - Monitoring of tumor dynamics	[83]
ERCC1 (Excision repair cross-complementation group 1)	NSCLC	Platinum chemotherapy	Correlation between low ERCC1 expression in CTCs and progression-free survival after platinum-based therapies	[85]
	Breast cancer	Neoadjuvant chemotherapy	- 72% of *ERCC1*-positive CTCs after therapy - No significant correlation between CTCs and clinical parameters	[86]
	Ovarian cancer	Platinum chemotherapy	*ERCC1*-positive CTC at diagnosis predictive of resistance to platinum-based therapy	[87]
SLFN11 (DNA/RNA helicase Schlafen family member 11)	CRPC	Platinum chemotherapy	Potential use of SLFN11 expression in CTCs for selection of patients with better response to platinum therapy	[92]
RAD23B (RAD23 homolog B)	Rectal cancer	Radiation and 5-FU Or radiation and capecitabine	Expression of thymidylate synthase (TYMS) and RAD23B has predictive value of nonresponse to neoadjuvant chemoradiation	[93]

## Data Availability

Not applicable.

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
