# Peer review of "Tumor Evolution and Therapeutic Choice Seen through a Prism of Circulating Tumor Cell Genomic Instability"

_cells, 2021, doi:10.3390/cells10020337_

Round 1

Reviewer 1 Report

In this review, the authors showed CTC genomic heterogeneity through FISH and single-cell sequencing. The authors highlighted the importance of genomic instability (GI) characterization in the context of tumor evolution and treatment choices. Additionally, the authors introduced the dynamic changes in DNA repair-related protein expression as a response to GI functional biomarker in CTC.

The manuscript introduced various GI-related CTC studies with recent references and well-described. If authors add one concept diagram for section2 (line 57-170), it will be easier for the reader to understand. 

Reviewer 2 Report

In this review, Tayoun and coauthors are aimed to emphasize the importance of assessment of genomic instability in CTCs for the study of tumor biology and prediction of therapy efficiency in cancer patients. Although this issue is not novel and covered in other reviews (e.g. Freitas et al., Cancers 2020), Tayoun et al. provided a more comprehensive and detailed analysis of the literature and their own data to support the idea of CTC applicability for tumor evolution tracing and therapeutic choice.

To make this review more clearly and close to clinical practice, I would like the authors to pay attention to the following points:

  1. The section “Genomic instability, more than a hallmark of cancer” is overloaded by the different facts and leads away from the main aim of the review. It can be reduced and focused on the basic information regarding GI and terminology.
  2. I doubt that monitoring of tumor evolution and therapeutic choice can be effective based on the analysis of CTCs. Indeed, CTCs are not a snapshot of the whole tumor but presented by tumor cells with a high potential for intravasation.
  3. CTCs are extremely heterogeneous in size, phenotype, and genotype. Therefore, cancer studies, especially focusing on tumor evolution and therapeutic choice, should be based on the analysis of all types/populations/subsets of CTCs. However, current methods allow isolating only certain CTCs, for example, which have large size (using microfluidics) or carry any marker (by Cell Search, FACS, MACS).
  4. CTCs are mostly detected in patients with advanced cancers and almost not observed at early stages. This fact significantly limits the category of patients for which analysis of CTCs can be applied.
  5. CTC-derived models are needed to study rather tumorigenic/metastatic potential of specific CTC populations and drug sensitivity. It is quite debatable that GI mechanisms should be investigated using these models. Anyway, the establishment of cell lines and xenografts from CTCs is highly challenging due to their low number in the blood of cancer patients.
  6. In my opinion, analysis of circulating tumor DNA/RNA is a more optimal instrument for tumor evolution tracing and prognosis of therapy response and cancer recurrence. It would be valuable if the authors could provide advantages and limitations of CTCs over ctDNA/RNA.

Round 2

Reviewer 2 Report

The authors provided comprehensive answers to my questions. Thank you.